# Comparison of Proanthocyanidin Content in Rabbiteye Blueberry (*Vaccinium virgatum* Aiton) Leaves and the Promotion of Apoptosis against HL-60 Promyelocytic Leukemia Cells Using ‘Kunisato 35 Gou’ Leaf Extract

**DOI:** 10.3390/plants12040948

**Published:** 2023-02-19

**Authors:** Yuki Toyama, Yoko Fujita, Saki Toshima, Tomonari Hirano, Masao Yamasaki, Hisato Kunitake

**Affiliations:** 1Graduate School of Agriculture, University of Miyazaki, 1-1 Gakuen-kibanadai nishi, Miyazaki 889-2192, Japan; 2Michimoto Foods Products Co., Ltd., 1667 Kou Tano-cho, Miyazaki 889-1701, Japan; 3Interdisciplinary Graduate School of Agriculture and Engineering, University of Miyazaki, 1-1 Gakuenkibanadainishi, Miyazaki 889-2192, Japan; 4Faculty of Agriculture, University of Miyazaki, 1-1 Gakuen-kibanadai nishi, Miyazaki 889-2192, Japan

**Keywords:** ‘Kunisato 35 Gou’, polyphenol, proanthocyanidin, HL-60, apoptosis

## Abstract

Polyphenol-rich rabbiteye blueberry (*Vaccinium virgatum* Aiton) leaves have attracted attention as a food material. In this study, we compared the total polyphenols, total proanthocyanidin content, and antioxidant activity of the leaves of 18 blueberry varieties and investigated the seasonal variation in polyphenols. We also evaluated the anti-cancer cell proliferation properties of the rabbiteye blueberry leaf specific cultivar ‘Kunisato 35 Gou’. Rabbiteye blueberry leaves had significantly higher total polyphenol and total proanthocyanidin values than northern highbush blueberry and southern highbush blueberry leaves. The antioxidant activity of blueberry leaves was highly positively correlated with both the total polyphenol and total proanthocyanidin content. Variations were observed in the total polyphenol and total proanthocyanidin content of rabbiteye blueberry leaves harvested at different points in the growing season; leaves collected in fall to winter contained more epicatechin in addition to proanthocyanidins. In the evaluation of anti-cancer cell proliferation properties against HL-60 promyelocytic leukemia cells, the September-harvested extracts of rabbiteye blueberry ‘Kunisato 35 Gou’ showed strong properties, and the use of an FITC Annexin V apoptosis detection kit with propidium iodide confirmed that this HL-60 cell death occurred via apoptosis. Limiting the harvest time would make rabbiteye blueberry leaves a more functional food ingredient.

## 1. Introduction

Blueberries (*Vaccinium* spp.) are a major crop worldwide. Blueberries are native to North America and classified into three species based on taxonomic differences: highbush blueberry (*V. corymbosum* L.), rabbiteye blueberry (*V. virgatum* Aiton), and lowbush blueberry (*V. angustifolium* Aiton) [1]. Lowbush blueberry is the wild type in North America, and highbush blueberry is suitable for cold areas. Conversely, rabbiteye blueberry is resistant to high temperatures and dryness in summer and is thus suitable for warm regions. The three species of blueberries have a great deal of phenotypic variability among them. As its cultivation has spread, the functionality of the blueberry fruit in the human diet has come to be a focus of attention [2,3]. The demand for blueberry plants has significantly increased in the last 30 years due to its market expansion. The blueberry production acreage is increasing yearly, as is the number of blueberry varieties. Health interest also continues to drive the demand for blueberries [4]. The blueberry size, taste, sugar acid ratio, and functionality, all of which differ among varieties, are being improved [5,6,7].

The blueberry fruit contains various types of polyphenols [8,9] and is popular for its richness in anthocyanins. Blueberry fruits contain delphinidin, cyanidin, petunidin, malvidin, and peonidin as anthocyanins [10,11], as well as gallic acid, ferulic acid, ellagic acid, catechin, epicatechin, rutin, chlorogenic acid, p-coumaric acid, caffeic acid, and quercetin as polyphenols [11,12,13]. Cancer prevention effects and improvements in lifestyle-related diseases from blueberries have been suggested [14,15]. The blueberry is considered to be a functional ingredient that maintains and promotes health and that is used as a raw material not only for processed foods but also for health-functional foods and pharmaceuticals. Blueberry leaves and stems, which traditionally have not been utilized, have now also become the subject of functional evaluation. Debnath-Canning et al. [16] showed that blueberry leaves had significantly higher total phenolic and flavonoid content than the fruit. Takeshita et al. [17] showed that oligomeric proanthocyanins from rabbiteye blueberry leaves suppressed the expression of hepatitis C virus (HCV) subgenomic RNA in an HCV replicon cell system. In parallel with this study, Yuji et al. [18] also examined the preventive effects of rabbiteye blueberry leaf extracts on lifestyle-related diseases such as lipid disorders, and the histopathological analysis of hepatic tissues in a rat model revealed that BL administration suppressed fatty infiltrations induced by an AIN-76-based high-sucrose diet. These authors suggested that dietary blueberry leaves may be useful for the prevention of fatty liver diseases. Along with these reports, rabbiteye blueberry extracts have exhibited numerous types of functional activities: anti-hypertensive [19], anti-atherosclerotic [20], and anti-diabetic activity [21]; anti-inflammatory properties [22]; insulin resistance [23]; and alcohol hypermetabolism [24]. Further, a recent human clinical study showed that beverages including rabbiteye blueberry extracts suppressed the postprandial increase in triacylglycerols [25]. Surprisingly, Ichikawa et al. [26] reported that the proanthocyanidin fraction in rabbiteye blueberry leaves had strong antiviral activity against hepatitis C virus (HCV) and human T-lymphocytic leukemia virus type 1 (HTLV-1). Moreover, a strong antiviral effect was observed in fraction 7, with high polymerized proanthocyanidin content in both the leaves and stems [27]. Blueberry leaves and stems are potential new materials for the prevention of various lifestyle-related and viral diseases.

Therefore, we intercrossed rabbiteye blueberry cultivars with high proanthocyanidin content with a variety with high proanthocyanidin content and high cutting propagation efficiency. We registered the resulting variety in 2014 as ‘Kunisato 35 Gou’, a blueberry-leaf-specific cultivar (No. 23433). The leaves are now being mass-produced by mechanical harvesting and other methods in Miyazaki Prefecture, Japan (Figure 1A–C), and the production of blueberry leaf tea has begun (Figure 1D). However, the polyphenol content and composition, as well as the antioxidant activity of rabbiteye blueberry leaves at different growth stages, have not yet been clarified.

In the present study, we investigated the total polyphenol content and composition, the proanthocyanidin content, and the antioxidant activity of the leaves of blueberry cultivars. We also evaluated the anti-cancer cell proliferation properties of the rabbiteye blueberry leaf specific cultivar ‘Kunisato 35 Gou’.

## 2. Results

### 2.1. Comparison of Total Polyphenol Content among 18 Varieties

The total polyphenol content of 18 varieties was estimated by the Folin–Ciocalteu reagent method, and the polyphenol content was expressed as mg gallic acid equivalents·g^−1^ DW (Figure 2). The total polyphenol content of rabbiteye blueberry leaves was significantly higher than that of southern and northern highbush blueberry leaves. Among rabbiteye blueberry leaves, the total polyphenol content of ‘Ethel’ (444 mg gallic acid equivalents·g^−1^ DW) was significantly higher than that of the other varieties. On the other hand, the total polyphenol content of ‘Magnolia’, ‘Capefear’, and ‘Gulfcoast’ (144, 149, and 168 mg gallic acid equivalents·g^−1^ DW, respectively) was significantly lower than that of southern and northern highbush blueberry leaves. Differences in the total polyphenol content of blueberry leaves were observed among varieties, and the content of the rabbiteye blueberry leaves tended to be significantly higher.

### 2.2. Comparison of Total Proanthocyanidin Content among 18 Varieties

The total proanthocyanidin content of the 18 varieties of blueberry was examined (Figure 3). Rabbiteye blueberry leaves had significantly higher content than southern and northern highbush blueberry leaves, as was the case with the total polyphenol content. In particular, the total proanthocyanidin content of ‘Gardenblue’ and ‘Woodard’ leaves (27.2 and 26.3 mg as catechin·g^−1^ DW, respectively) was significantly higher than that of the other 16 varieties. On the other hand, the total proanthocyanidin content of southern blueberry varieties ‘Magnolia’, ‘Capefear’, and ‘Gulfcoast’ (2.0, 2.2, and 3.2 mg gallic acid equivalents·g DW^−1^, respectively) was significantly lower than that of the other 15 varieties. The rabbiteye blueberry varieties had significantly higher total proanthocyanidin content, while many southern highbush blueberry leaves showed lower levels.

### 2.3. Antioxidant Activity Analysis

The antioxidant activity of 18 varieties was estimated by the DPPH free radical scavenging assay, and the values were expressed as µmol as Trolox·g^−1^ DW (Figure 4). The antioxidant activity of rabbiteye blueberry leaves was significantly higher than that of southern and northern highbush blueberry leaves. In rabbiteye blueberry leaves, the antioxidant activity of ‘Ethel’ and ‘Gardenblue’ (2363 and 2329 µmol as Trolox·g^−1^ DW, respectively) was higher than those of the other varieties. On the other hand, the antioxidant activity of ‘Magnolia’ (646 µmol as Trolox·g^−1^ DW) was lower than that of the other southern and northern highbush blueberry leaves. Differences in the antioxidant activity of blueberry leaves were observed among varieties, and the leaves of the rabbiteye blueberry tended to have significantly higher content, as was also the case with the total polyphenol content and the proanthocyanidin content.

### 2.4. Changes in the Total Polyphenol Content and the Total Proanthocyanidin Content of Rabbiteye Blueberry Leaves throughout the Growing Season

Leaves of rabbiteye blueberry throughout the growing seasons were tested for their total polyphenol content and total proanthocyanidin content (Figure 5A,B). Rabbiteye blueberry ‘Homebell’ had young leaves in the spring, and then immature leaves expanded and hardened for the summer and finally turned red and defoliated for the winter. The total polyphenol content decreased significantly at one point toward May but then increased rapidly toward October and stabilized toward December. On the other hand, the total proanthocyanidin content increased rapidly from April to September and then decreased slightly from October onward, but remained stable. In summary, the total polyphenol content and proanthocyanidin content of rabbiteye blueberry leaves varied significantly depending on the growth stage and were lower in spring and higher from summer to fall.

### 2.5. Polyphenol Components of Rabbiteye Blueberry Leaves throughout the Growing Season

The leaves of rabbiteye blueberry contained four main polyphenols throughout the growing season: chlorogenic acid, rutin, catechin, and epicatechin. The content of chlorogenic acid, rutin, catechin, and epicatechin in the leaves of rabbiteye blueberry ‘Homebell’ was investigated throughout the growing season (Figure 6). In the April leaves immediately after expanding, chlorogenic acid accounted for 31.2% of the polyphenols that could be detected (268.0 gallic acid equivalents·g^−1^ DW). As the leaves grew, the chlorogenic acid content declined rapidly at first and then slowly after October. The rutin levels were not as high as those of chlorogenic acid but were high in April and then declined moderately. On the other hand, the epicatechin content of leaves was very low in April but increased rapidly after July and continued to increase until December, shortly before the leaves fell. The epicatechin content of leaves in December accounted for around 26.7% of the polyphenols detected (311.7 gallic acid equivalents·g^−1^ DW). Catechins were not high as a whole but increased moderately from September onward when the epicatechin content increased. As noted above, the polyphenolic composition of rabbiteye blueberry leaves changed dramatically as the leaves grew, with different major polyphenols.

### 2.6. Anti-Cancer Cell Proliferation Properties against HL-60 Cells

Figure 7A shows changes in the relative viability of the HL-60 cells (i.e., survival rate) by treatment with leaf extracts of ‘Homebell’ (April) and ‘Kunisato 35 Gou’ (April and September). In April, there were no significant differences in cell viability between the two treatments, although the survival rate tended to increase as the concentration decreased. In September, on the other hand, when treated with 5.0 mg·mL^−1^ ‘Kunisato 35 Gou’, there were no viable HL-60 cells. The survival rate tended to increase as the concentration of leaf extract decreased.

### 2.7. Anti-Cancer Cell Proliferation Properties against HL-60 Cells

To detect the ratio of early/late apoptosis induced by the treatment of HL-60 cells with a concentration of 0.625 mg·mL^−1^ of each extract, the cells were labelled by FITC Annexin V and propidium iodide solution (Figure 7B). In a comparison of the early/late apoptosis rate in April leaf extracts of rabbiteye blueberry ‘Homebell’ and ‘Kunisato 35 Gou’, the latter resulted in a higher rate. The early/late apoptosis rate of leaf extract of ‘Kunisato 35 Gou’ (September) was significantly higher than those of the others. Moreover, the percentages of both early and late apoptotic cells were roughly the same.

## 3. Discussion

Polyphenols are secondary compounds that are widely distributed in plants and are classified into four groups according to their diverse chemical structures: flavonoids, lignans, stilbenes, and phenolic acids [28]. These polyphenols in plants play important roles in protecting the organism from external stimuli and in removing reactive oxygen species that cause various diseases [29]. Because of their high levels of antioxidant activity, these compounds have been extensively studied in recent years for their beneficial nutritional and health effects [30,31,32]. The polyphenol accumulation and profiles of plant leaves, which are an important source of polyphenols, are influenced by seasonal climatic conditions, biotic and abiotic stressors, soil, cultural practices, and genetics [33]. Our research group reported that blueberry leaves are a rich source of polyphenols and have many health benefits [25,26]. However, genotypic and seasonal changes in the content and composition of polyphenols that affect the quality of blueberry leaves have not been examined to date. In this study, we characterized the polyphenols and antioxidant activity of the leaves of many blueberry cultivars and investigated the inhibitory effects of their extracts on cancer cell proliferation via apoptosis.

Differences in the total polyphenol content and the total proanthocyanidin content of blueberry leaves were observed among species and strains, and, in particular, the group of rabbiteye blueberry cultivars had significantly higher values of them compared to the other species and strains. A similar trend was observed in the evaluation of the antioxidant ability. The antioxidant ability of blueberry leaves among 18 varieties showed high positive correlations with both the total polyphenol content (*r* = 0.9885) and the total proanthocyanidin content (*r* = 0.9574). These results suggest that polyphenols, particularly proanthocyanidins, likely contribute to the antioxidant ability of blueberry leaves.

The polymerized proanthocyanidin structure of rabbiteye blueberry leaves consists mainly of B-type bonds, but there are also type A bonds and cinchonain I units [34]. This unique structure and the degree of polymerization also affect their health benefits [27,35,36], and various functions have been reported, such as inhibiting hepatitis C virus replication [17], suppressing fatty liver development [37], inhibiting the progression of hepatocarcinogenesis [38], suppressing the development of adult T-cell leukemia [39], and inhibiting the proliferation of HTLV-1 virus-infected cells [40].

Wang and Nambeesan [41] reported that rabbiteye blueberries were native to North America and that breeding efforts to improve blueberry fruit quality focus on improving specific traits, such as increased firmness, enhanced flavor, and greater shelf-life. That is, highbush blueberries originated from the cold region of northeastern North America, and southern highbush blueberries were derived from crosses between highbush blueberries and *Vaccinium* species native to Florida, in the southern United States (including *V. darrowii* Camp.) [42]. Considering species and strain considerations, it is likely that the rabbiteye blueberry evolved to accumulate high polyphenol levels in its leaves out of a need to protect itself from adverse environmental conditions such as high temperatures and UV light.

The total polyphenol and proanthocyanidin content of rabbiteye blueberry leaves was low in spring immediately after leaf expansion and gradually increased toward summer, when the temperature and light intensity were higher. In the polyphenolic composition of rabbiteye blueberry leaves, the chlorogenic acid content was highest in April leaves and decreased sharply after May. On the other hand, the epicatechin content was low in spring leaves but increased rapidly and was highest in December leaves. Sorbus domestica L. leaves are also a phenylpropanoid-rich herbal medicine with antioxidant and anti-inflammatory activities [43]. Authors reported similar seasonal dynamics between the total proanthocyanidin content and total polyphenol content of leaves, meaning the highest content of both was observed in late summer and fall. These results are consistent with seasonal variations in the composition of rabbiteye blueberry leaves. The production and accumulation of secondary metabolites in plants are species-specific and tightly regulated [44]; they are controlled by complex regulatory mechanisms and respond precisely to stimulating environmental factors, including those related to seasonal changes [45]. Investigating seasonal variations in functional components of blueberry leaves, such as the proanthocyanidin content, and the individual polyphenol composition and content of blueberry leaves may be an important indicator for harvesting quality material.

In the evaluation of the anti-proliferation properties of blueberry leaf extracts against HL-60 cells, the September leaf extracts of rabbiteye blueberry ‘Kunisato 35 Gou’ showed strong activity that was slightly stronger than that of its parent, ‘Homebell’. In the apoptosis experiment using the FITC Annexin V apoptosis detection kit with propidium iodide, these properties were shown to be mediated via apoptosis, and the sum of the early and late apoptosis of ‘Kunisato 35 Gou’ leaves in September was the highest. We speculated that polyphenols such as proanthocyanidins and epicatechin contribute to this high apoptosis-mediated cancer cell suppression effect. Previous studies have also shown that several polyphenols and proanthocyanidins induce apoptosis-mediated anti-cancer cell proliferation properties [46,47,48,49]. Ahmadi et al. [50] reported that epicatechin- and scopoletin-rich Morinda citrifolia leaf extracts ameliorated leukemia via cancer cell apoptosis. Song et al. [49] described that the proanthocyanidins isolated from the leaves of Photinia × fraseri could affect melanin production by downregulating microphthalmia transcription factor expression and inhibiting the activity of tyrosinase and tyrosinase-related protein 1, leading to cell cycle arrest and the apoptosis of melanoma cells. As can be understood from these reports, proanthocyanidin- and epicatechin-rich rabbiteye blueberry ‘Kunisato 35 Gou’ leaves were found to be a food material with high anti-cancer cell proliferation properties.

The present study revealed that the leaves of rabbiteye blueberries have higher total polyphenol content, higher total proanthocyanidin content, and stronger antioxidant activity than other species and hybrids. In addition, significant seasonal variation was observed in the polyphenol and proanthocyanidin content of rabbiteye blueberry leaves. Moreover, the September leaf extracts of rabbiteye blueberry leaf specific cultivar ‘Kunisato 35 Gou’ showed high apoptosis-mediated anti-cancer cell proliferation properties. Recently, the health-beneficial function of ‘Kunisato 35 Gou’ was shown to depend on the chemical structure and degree of polymerization in the case of proanthocyanidins [27,36]. In the future, it will be necessary to investigate how different proanthocyanidin structures and degrees of polymerization affect the health benefits.

## 4. Materials and Methods

### 4.1. Plant Materials

Blueberry plants were cultivated for more than 5 years in a field at the Faculty of Agriculture at the University of Miyazaki (31°49′41.2″ N 131°24′41.0″ E). Plants were spaced at 3 m between rows on raised beds covered with strips of nonwoven fabric for nurseries, which protected the plants from weeds. Fundamental cultivation methods were carried out according to the guidelines for cultivation in Miyazaki Prefecture. Mature leaves of 10 varieties of rabbiteye blueberries (‘Climax’, ‘Swannee’, ‘Meyers’, ‘Callaway’, ‘Ethel’, ‘Southland’, ‘Blue-belle’, ‘Gardenblue’, and ‘Woodard’), 5 varieties of southern highbush blueberries (‘Magnolia’, ‘Gulfcoast’, ‘Capefear’, ‘Hardyblue’, and ‘Sunshineblue’), and 4 varieties of northern highbush blueberries (‘Spartan’, ‘Shef’, ‘Berkeley’, and ‘Darrow’) were used to investigate the total polyphenol content, total proanthocyanidin content, and antioxidant activity. Leaves were harvested when the leaves were completely matured in June and July.

Then, seasonal changes in total proanthocyanidins, total polyphenol content, and composition, and the anti-cancer cell proliferation properties, were examined using mature leaves harvested from blueberry leaves cultivated using improved Japanese green tea cultivation techniques. The rabbiteye blueberry varieties ‘Homebell’ and ‘Kunisato 35 Gou’ with high polyphenol content were used to assess the seasonal changes in functional ingredients and the anti-cancer cell proliferation properties, respectively. In the field that we used for our study, the width between rows was 0.3 m, and the canopy width of the plants was 1.2 m (Figure 1A). These densely spaced plants were pruned each October for 3 years, and mature leaves were sampled 4 years after planting and harvested once a month from April to December.

The collected samples were immediately frozen and then dried in a freeze dryer (FDU-2100, EYELA, Tokyo, Japan) and a square dry chamber (DRC-1000, EYELA, Tokyo, Japan). These dried samples were crushed using a crusher (BUCHI Mixer B-400, BUCHI, Tokyo, Japan) and then stored in a freezer at −20 °C.

### 4.2. Polyphenol Analysis

For polyphenol analysis, each sample was prepared by dissolving freeze-dried leaf powder (0.02 g) in 5 mL of 80% (*v*/*v*) methanol and passing it through a 0.22 µm membrane filter (Millipore, Bedford, MA, USA). The total polyphenol content was measured according to the Folin–Ciocalteu reagent method [51]. Briefly, 200 μL of each sample, 200 µL of phenol reagent (distilled water 5.0 mL and Folin–Ciocalteu’s phenol reagent), and 400 µL of saturated sodium carbonate solution were mixed. Absorbance was read at 760 nm after 30 min. Gallic acid dissolved in 80% (*v*/*v*) methanol was used as the standard. The total polyphenol content was expressed as g gallic acid equivalents·100 g^−1^ DW. Each sample extract was measured three times.

The polyphenol components of leaves were clarified by separating and identifying individual polyphenols by HPLC. To prepare leaf samples, 0.02 g of frozen berries was extracted with 5 mL of 80% (*v*/*v*) methanol. The extraction was performed by an ultrasonic device (US CLEANER, As One, Osaka, Japan) for 15 min at 37 °C and samples were passed through a 0.22 µm membrane filter (Millipore) for HPLC analysis, which was performed using a Prominence LC solution system (Shimadzu, Kyoto, Japan) with Inertsil ODS3 (Shimadzu) (4.6 mm × 250 mm, 5 µm). The chromatographic conditions were as follows: solvent A, 100% (*v*/*v*) ethanol; solvent B, 20 mM potassium phosphate (pH 2.4); column temperature, 40 °C; detection at 280 nm; flow rate, 1.0 mL·min^−1^. The binary gradients were as follows: 85–68% B (0–12 min), 68% B (12–15 min), 68–55% B (15–20 min), 55–85% B (20 min), and 85% (20–29 min). Retention times and spectra were compared with pure standards of chlorogenic acid, catechin, epicatechin, rutin, and caffeic acid. These extracts were diluted within the range of 20 to 200 mg·L^−1^ by 100% (*v*/*v*) methanol. The concentrations were designed to yield comparable peak heights for the easy derivation of chromatographic parameters. The peak areas of the standard and the sample were normalized and used for the quantitation of the active ingredients, whose content was subsequently expressed as a percentage of the label claim. The results were expressed as g gallic acid equivalents·mg·100g^−1^ DW. Each sample extract was measured three times.

### 4.3. Total Proanthocyanidin Analysis

The same materials used for polyphenol analysis were used for the analysis of total proanthocyanidin content. The p-dimethylaminocinnamaldehyde (DMACA) method was used for total proanthocyanidin content. Briefly, hot water extracts were prepared by the same method used for the polyphenol analysis. Then, the appropriately diluted extract sample and standard solution were added. After mixing 0.1% DMACA and allowing the admixture to stand for 20 min, the absorbance at 640 nm was measured; 1N HCl in methanol was used for the blank. In the standard solution, catechin was dissolved in methanol. The standard curve was linear between 2.5 and 40 µg·mL^−1^. To calculate the total amount of proanthocyanidins, after subtracting the absorbance value of the reagent blank from the absorbance value of each well, we plotted the concentration (mg/L) of the standard solution on the *X*-axis and the absorbance value of the standard solution on the *Y*-axis. A calibration curve was created, and the total proanthocyanidins in the sample were calculated in terms of (+)-catechin equivalent according to the following formula.
$$Wr=\frac{\left(A_{\mathrm{sumple}}-A_{\mathrm{intercept}}\right)\times V\times d}{S_{\mathrm{std}}\times m}\;$$

*W*r: total proanthocyanidin content (mg/g), *A*_sample_: absorption value of measurement sample solution, *A*_intercept_: absorbance value of standard curve *Y*-axis intercept, *V*: sample extract volume, *d*: sample dilution ratio, *S*_std_: slope of calibration curve, *m*: sample amount.

### 4.4. Antioxidant Activity

Each sample was prepared by dissolving freeze-dried leaf powder (0.1 g) in 80% (*v*/*v*) ethanol and passing it through a 0.45 µm membrane filter (Millipore, Bedford, MA, USA). The antioxidant activity was determined by a 1,1-diphenyl-2-picrylhydrazyl (DPPH) free radical scavenging assay according to the method described by [52]. In brief, 50 µL of 20% ethanol solution and 50 µL of a 200 mM 2-morpholinoethanesulphonic acid buffer (pH 6.0) were added to 50 µL of sample solution per well of a 96-well microplate. The reaction was started by the addition of 50 µL of 1.2 mM DPPH to ethanol. Twenty minutes after these were added at room temperature, the absorbance at 520 nm was measured using an Immuno-Mini NJ-2300 microplate reader (Nalge Nunc International, Tokyo, Japan) with Trolox (Aldrich Japan, Tokyo, Japan). Trolox was used as a standard. The antioxidant activity is expressed as µmol Trolox·g^−1^ DW. Each sample extract was measured three times.

### 4.5. Cell Line Used for the Analysis of the Anti-Cancer Cell Proliferation Properties and Flow Cytometric Propidium Iodide Annexin V Assay

The HL-60 (human promyelocytic leukemia cells) cell line was used. Cells at a concentration of 1.0 × 10^5^ cells·mL^−1^ were incubated in an air atmosphere humidified with 5% CO_2_ for 24 h at 37 °C in a CO_2_ incubator (Thermo Fisher Scientific, Waltham, MA, USA). HL-60 cells were cultured with RPMI-1640 medium (Fujifilm Wako, Osaka, Japan) supplemented with 10% fetal bovine serum containing 100 units·mL^−1^ penicillin and 100 µg·mL^−1^ streptomycin.

### 4.6. Preparation of Extracts for Analysis of Anti-Cancer Cell Proliferation Properties

Anti-cancer cell proliferation properties (survival rates) were measured according to the Technical Manual of Cell Counting Kit-8 (Dojindo Molecular Technologies, Inc., Kumamoto, Japan). Freeze-dried leaf powders (0.5 g) were homogenized in 20 mL of 80% (*v*/*v*) methanol. The extracts (5.0 mg·mL^−1^) were dissolved by an ultrasonic cleaner (Aswan, Osaka, Japan), and centrifugal separation was performed. The supernatants were filtered through a 0.22 µm membrane filter and concentrated by rotary evaporation at 40 °C. The extracts were then lyophilized and tested as leaf methanol extracts. The extracts were dissolved in dimethyl sulfoxide (DMSO) to 100 mg·mL^−1^ and then diluted with medium to make seven concentrations (5, 2.5, 1.25, 0.625, 0.3125, and 0.156 µg·mL^−1^). Each concentration of extract was added to a cell suspension and incubated for 24 h. Then, 50 µL of 10-fold diluted CCK-8 solution was added to each well. Absorbance was measured after 1 h. The calculation was performed according to the protocol (Dojindo Molecular Technologies, Inc., Kumamoto, Japan), and the relative viable cell number as the survival rate was determined for each concentration and expressed as a percentage of 100% control [53].

### 4.7. Flow Cytometric Propidium Iodide Annexin V Assay

Annexin straining (FITC Annexin V Apoptosis Detection Kit with propidium iodide, BioLegend, San Diego, CA, USA) was performed to investigate the rate of apoptosis. The extracts were used to assess anti-cancer cell proliferation properties, especially for the 0.625 mg·mL^−1^ extracts. Each concentration of extract was added to a cell suspension and incubated for 24 h. The assay was performed according to the instructions of the kit. Briefly, cultured cells with samples were rinsed with phosphate-buffered saline for 24 h. Then, cells with binding buffer at a concentration of 4.0 × 10^6^ cells·mL^−1^ were transferred to 100 µL of suspension in a 5 mL test tube. Finally, 5 μL and 10 μL, respectively, of FITC Annexin V and propidium iodide solution were added. When measuring, depending on the fluorescence intensity of Annexin V and propidium iodide, the populations could be distinguished into double-negative (healthy), Annexin-V-positive (early apoptotic cells), and double-positive (late apoptotic or necroptotic) cells. As a control, only cells were used without the addition of a sample.

### 4.8. Statistical Analysis

All experimental results were obtained from triplicate measurements (three extracts and three measurements per extract), and the data in the tables and figures represent mean values ± standard deviation (n = 3). The results were evaluated for statistical significance at *p* < 0.05 using univariate analysis of variance (ANOVA) and Tukey’s post hoc test. We investigated the correlations among total anthocyanin content, total polyphenol content, antioxidant activity (DPPH), and anti-cancer cell proliferation properties by CORREL (Correlation in Excel).

## Figures and Tables

**Figure 1 plants-12-00948-f001:**
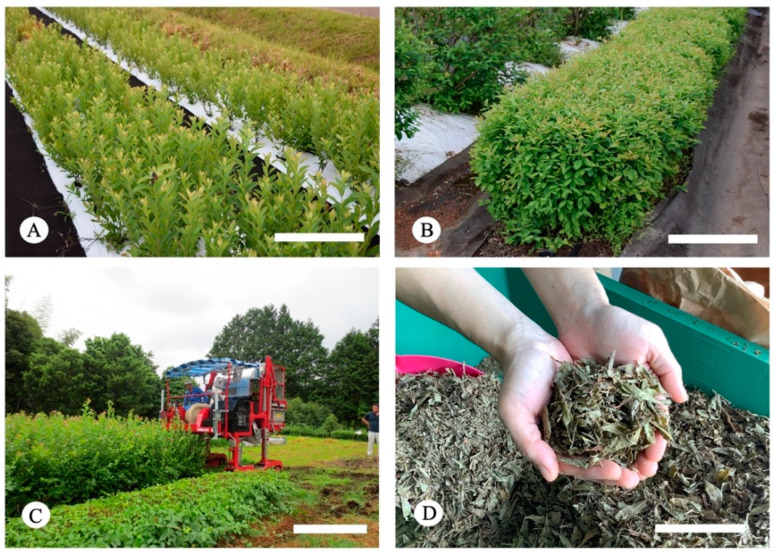
Mass production of leaves and teamaking from rabbiteye blueberry ‘Kunisato 35 Gou’. (**A**) Dense planting of cutting nursery. Bar = 50 cm. (**B**) Leaf production method similar to Japanese green tea cultivation. Bar = 50 cm. (**C**) Mechanical harvesting. Bar = 100 cm. (**D**) Dried tea leaves. Bar = 10 cm.

**Figure 2 plants-12-00948-f002:**
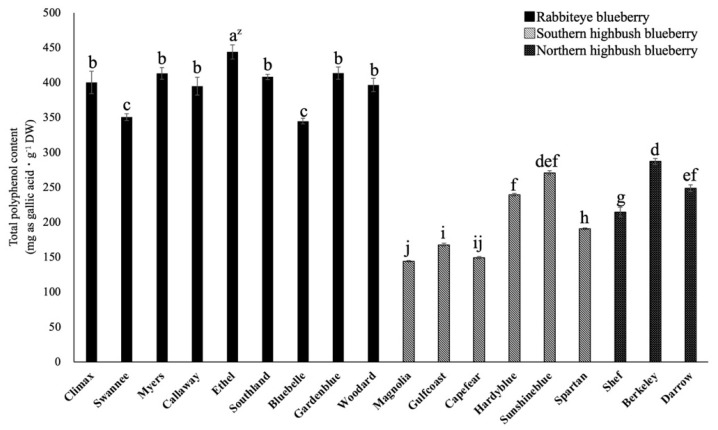
Comparison of the total polyphenol content in mature leaves of blueberry varieties. ^z^ Different letters among varieties represent significant differences at 5% level as determined by Tukey’s multiple range test (n = 3).

**Figure 3 plants-12-00948-f003:**
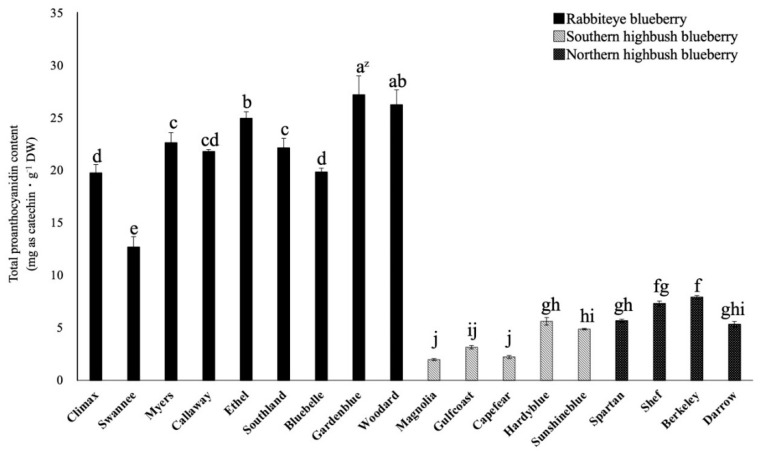
Comparison of the total proanthocyanidin content in mature leaves of blueberry varieties. ^z^ Different letters among varieties represent significant differences at 5% level as determined by Tukey’s multiple range test (n = 3).

**Figure 4 plants-12-00948-f004:**
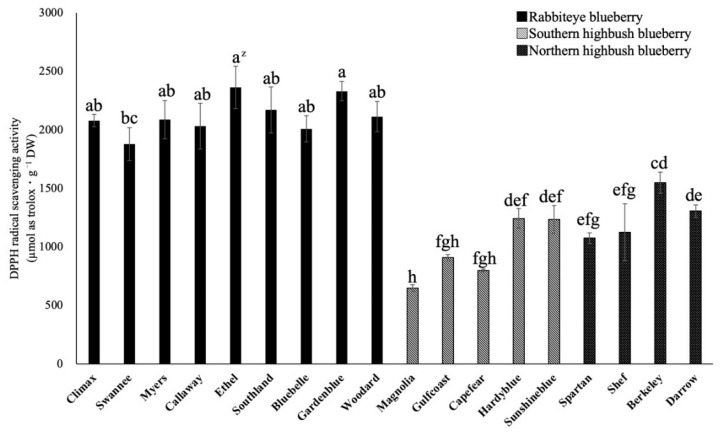
Comparison of the antioxidant activity in mature leaves of blueberry varieties. ^z^ Different letters among varieties represent significant differences at 5% level as determined by Tukey’s multiple range test (n = 3).

**Figure 5 plants-12-00948-f005:**
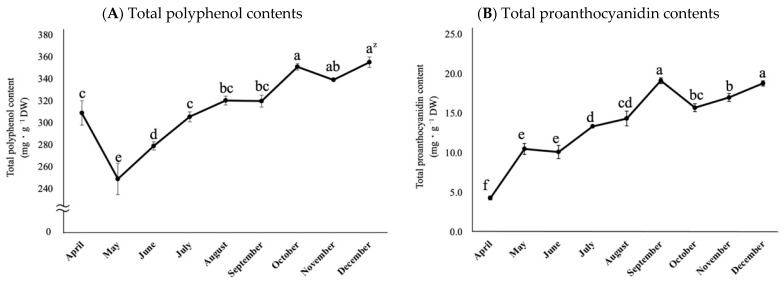
Seasonal changes in the polyphenol content (**A**) and polyphenol content (**B**) in mature leaves of rabbiteye blueberry Homebell. ^z^ Different letters among months represent significant differences at 5% level as determined by Tukey’s multiple range test (n = 3).

**Figure 6 plants-12-00948-f006:**
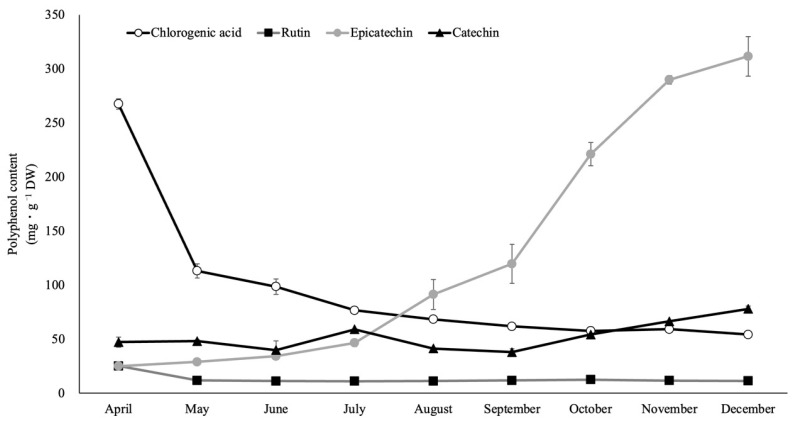
Seasonal changes in content of chlorogenic acid, rutin, catechin, and epicatechin in mature leaves of rabbiteye blueberry ‘Homebell’.

**Figure 7 plants-12-00948-f007:**
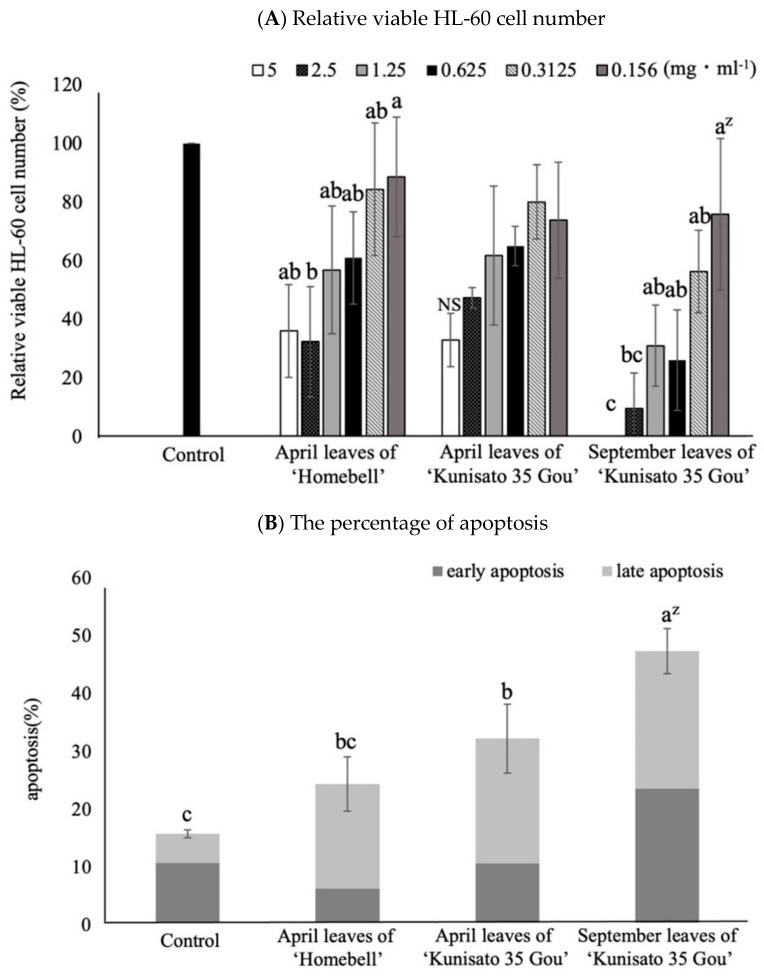
Effects of blueberry leaf extracts on the anti-cancer cell proliferation properties and apoptosis in HL-60 cells (human promyelocytic leukemia cells). (**A**). Comparison of the relative viable HL-60 cell numbers (survival rates) against leaf extracts of ‘Homebell’ in April and ‘Kunisato 35 Gou’ in April and September. Only HL-60 cells were used without adding extract as a control. ^z^ Different letters among 6 concentrations represent significant differences at 5% level as determined by Tukey’s multiple range test (n = 3). The relative viable cell number was determined and expressed as % of the viability of cells of the control, which was determined as 100%. (**B**) The early/late apoptosis rate against leaf extracts of rabbiteye blueberry ‘Homebell’ in April and ‘Kunisato 35 Gou’ in April and September. ^z^ Different letters among varieties each month represent significant differences at 5% level as determined by Tukey’s multiple range test (n = 3). Only HL-60 cells were used without adding extract as a control.

## Data Availability

Not applicable.

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
