# Peer review of "Comparison of Proanthocyanidin Content in Rabbiteye Blueberry (Vaccinium virgatum Aiton) Leaves and the Promotion of Apoptosis against HL-60 Promyelocytic Leukemia Cells Using ‘Kunisato 35 Gou’ Leaf Extract"

_plants, 2023, doi:10.3390/plants12040948_

Round 1

Reviewer 1 Report

Dear Author, 

In my opinion the manuscript is a fair and useful study in line to the aim of the journal. The design is well executed and the manuscript is very well written. I have only few minor suggestions to make as presented below. 

Row 392 Please indicate the cell origin. Please indicate why leukemia cells were selected from the wide range of cancers?

Row 432 to be relevant p should be < 0.05, not = 0.05

Row 433 please mention what correlation test did you use… ANOVA is not a correlation test. Because in the methods no correlation results were described I suggest replacing the word correlation with something else…  

Author Response

Row 392: Please indicate the cell origin. Please indicate why leukemia cells were selected from the wide range of cancers?

Answer: Thank you for the suggestion. We chose leukemia cells because many papers reported anticancer with treatment of fruit extract by using The HL-60 (human promyelocytic leukemia cells) cell line.   (Maher et al, 2021, Title: Medicinal Plants with Anti-Leukemic Effects: A Review, DOI: https://doi.org/10.3390/molecules26092741)

Row 432 to be relevant p should be < 0.05, not = 0.05

Answer: Thank you for your advice. I changed it.

Row 433 :please mention what correlation test did you use… ANOVA is not a correlation test. Because in the methods no correlation results were described I suggest replacing the word correlation with something else…  

Answer: Thank you for your advice. I added the ‘CORREL in Excel’ in the sentense.

Reviewer 2 Report

The publication presented for review: „Comparison of proanthocyanidin content in rabbiteye blueberry …” is very interesting and constitutes an interesting research area. The publication is written very precisely and the issues are presented in detail, among others, cultivation, harvesting, as well as the methodology of testing the contents of polyphenols and proanthocyanidin. Noteworthy is the fact of research on the anticancer and antioxidant activities of rabbiteye blueberry extracts described in the publication. `Kunisato 35 Gou` (cultivar of the rabbiteye blueberry) showed high anticancer cell proliferation properties. I have only one small remark. I think that an exemplary proanthocyanidin formula could be included in the publication.

Author Response

Thank you for your advice. I added the calculation method of proanthocyanidin analysis (Materials and Methods, Line 382383. Fuethermore, the characters of proanthocyanidin formula described in Discussion (Line 252-253).

The polymerized proanthocyanidin structure of rabbiteye blueberry leaves consists mainly of B-type bonds but there are also type A bonds and cinchonain I units [34].

  1. Matsuo, Y.; Fujita, Y.; Ohnishi, S.; Tanaka, T.; Hirabaru, H.; Kai, T.; Sakaida, H.; Nishizono, S.; Kouno, I. Chemical constituents of the leaves of rabbiteye blueberry (Vaccinium ashei) and characterisation of polymeric proanthocyanidins containing phe-nylpropanoid units and A-type linkages. Food Chem. 2010, 121, (4), 1073-1079.

Reviewer 3 Report

English very difficult to understand and need improved.

there are many mistakes in Ms, such as 2.6 and 2.7 

Author Response

Thank you for pointing this out. We have made the following corrections and will revise the entire manuscript.  (https://www.mdpi.com/authors/english.)

2.6. Anti-cancer cell proliferation properties against HL-60 cells

Figure 7A shows changes in the relative viability of the HL-60 cell (i.e., survival rate) by the treatments with leaf extracts of ‘Homebell’ (April) and ‘Kunisato 35 Gou’ (April and September). In April, there were no significant differences in cell viability between the two treatments, although the survival rate tended to increase as the concentration decreased. In September, on the other hand, when treated with 5.0 mg⋅mL-1 ‘Kunisato 35 Gou’, there was no viable HL-60 cells. The survival rate tended to increase as the concentration of leaf extract decreased.

2.7. Anti-cancer cell proliferation properties against HL-60 cells

To detect the ratio of early/late apoptosis induced by the treatment of HL-60 cells with a concentration of 0.625 mg⋅mL-1 of each extract, the cells were labelled by FITC annexin V and propidium iodide solution (Figure 7B). In a comparison of the early/late apoptosis rate in April leaf extracts of rabbiteye blueberry ‘Homebell’ and ‘Kunisato 35 Gou’, the latter resulted in a higher rate. The early/late apoptosis rate of leaf extract of ‘Kunisato 35 Gou’(September) was significantly higher than those of the others. And the percentages of both early and late apoptotic cells were about the same.

Round 2

Reviewer 3 Report

accept